# Insights on Salt Tolerance of Two Endemic *Limonium* Species from Spain

**DOI:** 10.3390/metabo9120294

**Published:** 2019-11-29

**Authors:** Sara González-Orenga, P. Pablo Ferrer-Gallego, Emilio Laguna, M. Pilar López-Gresa, Maria P. Donat-Torres, Mercedes Verdeguer, Oscar Vicente, Monica Boscaiu

**Affiliations:** 1Instituto Agroforestal Mediterráneo (IAM), Universitat Politècnica de València, Camino de Vera s/n, 46022 Valencia, Spain; sagonor@doctor.upv.es (S.G.-O.); merversa@doctor.upv.es (M.V.); 2Centro para la Investigación y Experimentación Forestal, CIEF-Wildlife Service, Generalitat Valenciana, Avda Comarques del País Valencia,114, 46930 Quart de Poblet, Valencia, Spain; flora.cief@gva.es (P.P.F.-G.); laguna_emi@gva.es (E.L.); 3Instituto de Biología Molecular y Celular de Plantas (IBMCP), Universitat Politècnica de València - Consejo Superior de Investigaciones Científicas (CSIC), Camino de Vera s/n, 46022 Valencia, Spain; mplopez@ceqa.upv.es; 4Instituto de Investigación para la Gestión Integrada de Zonas Costeras. (IGIC), Universidad Politècnica de València. C/ Paranimf 1, 46730 Gandia, Valencia, Spain; mpdonat@eaf.upv.es; 5Instituto de Conservación y Mejora de la Agrodiversidad Valenciana (COMAV), Universitat Politècnica de València, Camino de Vera s/n, 46022 Valencia, Spain; ovicente@upvnet.upv.es

**Keywords:** *Limonium albuferae*, *Limonium dufourii*, growth parameters, photosynthetic pigments, ionic homeostasis, metabolic profiles, carbohydrates, organic acids, amino acids, multivariate analysis

## Abstract

We have analysed the salt tolerance of two endemic halophytes of the genus *Limonium*, with high conservation value. In the present study, seed germination and growth parameters as well as different biomarkers—photosynthetic pigments, mono and divalent ion contents—associated to salt stress were evaluated in response to high levels of NaCl. The study was completed with an untargeted metabolomics analysis of the primary compounds including carbohydrates, phosphoric and organic acids, and amino acids, identified by using a gas chromatography and mass spectrometry platform. *Limonium albuferae* proved to be more salt-tolerant than *L. doufourii*, both at the germination stage and during vegetative growth. The degradation of photosynthetic pigments and the increase of Na^+^/K^+^ ratio under salt stress were more accentuated in the less tolerant second species. The metabolomics analysis unravelled several differences between the two species. The higher salt tolerance of *L. albuferae* may rely on its specific accumulation of fructose and glucose under high salinity conditions, the first considered as a major osmolyte in this genus. In addition, *L. albuferae* showed steady levels of citric and malic acids, whereas the glutamate family pathway was strongly activated under stress in both species, leading to the accumulation of proline (Pro) and γ-aminobutyric acid (GABA).

## 1. Introduction

Salt-affected habitats house plant species with particular adaptations to withstand the deleterious effects of high salt concentrations in the soil, which may include the synthesis of specific metabolites [1,2]. Only halophytes, or salt-tolerant species, belonging to different plant families, are well adapted to survive in saline ecosystems; a good example is the *Plumbaginaceae* family, which consists mainly of perennial shrubs, subshrubs, and herbs, most of them growing in arid and saline habitats [3,4]; some of these species, mainly belonging to the genera *Limonium* Mill., *Limoniastrum* Fabr. and *Plumbago* L., are used as ornamental and medicinal plants.

The Mediterranean basin is one of the regions in the world with the greatest plant diversity [5] and is particularly rich in saline habitats. Within this region, the Spanish territory of the Iberian Peninsula and the Balearic Islands contain several major hotspots of plant diversity [6,7]; besides, this area holds an extensive list of endemic and threatened species [8,9]. *Limonium* is the genus with the highest number of recognised species in Spain [10,11,12,13,14], hosting not less than 107 species [15]. *Limonium* is also one of the genera with more conservationist relevance in Spain, including 74 species listed in the Spanish Red List of Threatened Vascular Plants [8], according to IUCN (International Union for Conservation of Nature) Criteria [16].

One of the Spanish territories richest in *Limonium* species is the Valencian Community, in which at least 26 endemic species are present, many of them listed as threatened [9,17,18,19,20,21]. The area called El Saler or Devesa de l’Albufera, near the city of Valencia, is one of the best known botanical sites in Spain, as well as the classical site for two of the most relevant Valencian *Limonium* endemics, *L. dufourii* (Girard) Kuntze, and *L. albuferae* P.P. Ferrer, R.A. Rosello and E. Laguna [17,18,19,20,21,22,23,24,25]. It is located in the l’Albufera Natural Park, the most renowned protected area of the Valencian Community. So far, little research has been conducted on these two endemic species and none specifically on their metabolite profiles.

*Limonium dufourii* is a triploid species (2n = 27) with obligate apomictic reproduction and incompatible pollen-stigma interactions [15,26,27]. It grows on sea-cliffs and salt marshes on sandy soils [17,18,20,23]. Although this species was more widely distributed in the past along the Valencian coasts and salt marshes, its current distribution is restricted to four populations, hosting from 232 (in the year 2004) to 36,435 (in the year 2011) individuals [28]. However, molecular analyses show that there is substantial genetic variability and differentiation within and between populations [27,29].

*Limonium albuferae* is a triploid (2n = 26), recently described as apomictic species with incompatible pollen-stigma interactions [25]. It is a perennial halophyte, multi-rosulate chamaephyte, varying from isolated rosettes (5–8 cm in width) to dense cushion-shape individuals (up to 45 cm in width) bearing tall flowering stems up to 80 cm (personal observations). *Limonium albuferae* has been found so far only in two small sites, both of them located in Devesa de l’Albufera. Despite the significant interest of these two threatened, endemic *Limonium* species for biodiversity conservation, there are no reports regarding their germination, salt tolerance and metabolite profiles. Seed germination and seedling establishment often represent the bottleneck of survival in saline environments [30,31]. Germination of salt marsh halophytes occurs when soil salinity is alleviated during the rainy season [32,33,34,35], which is generally autumn in the Mediterranean façade of the Iberian Peninsula [36,37].

Plant growth under stressful conditions depends on the efficiency of the mechanisms of stress tolerance of each particular species. A distinctive trait of many halophytes is their ability to absorb toxic ions from saline soils, transport them to the aerial part of the plant and sequester them into vacuoles of the foliar tissue, thus ensuring a low cytosolic concentration; this response is especially efficient in highly salt-tolerant dicotyledonous halophytes [38]. In parallel to mineral ion compartmentalisation in vacuoles, plants accumulate in the cytoplasm non-toxic, compatible solutes necessary for osmotic balance. Such compounds, known as osmolytes, are not specific for halophytes, but are also synthesised by glycophytes under different abiotic stress conditions that cause depletion of cellular water, such as drought, high salinity in the soil or extreme temperatures [39,40]. A wide range of metabolites are involved in responses to salinity, including mono-, di-, oligo- and polysaccharides, sugar alcohols, amino acids, quaternary ammonium compounds, betaines and tertiary sulphonium compounds [40]. Sugars are direct products of photosynthesis that play diverse essential functions in plant cells, so that the assessment of their specific roles as compatible solutes is not so simple. An increase in their concentration is not necessarily a primary response to stress, but could be due to activation of other cellular processes [41]. The major roles played by soluble carbohydrates in stress mitigation involve osmoprotection, carbon storage, and scavenging of reactive oxygen species (ROS) [40]. Amino acids are the constituents of proteins, but they have also regulatory and signalling functions. Many amino acids have a well-established role in stress tolerance, such as the flagship compatible solute Pro. Recent microarray studies and amino acids profiles in stressed *Arabidopsis thaliana* plants indicated that an increase in their contents under stress is a general trend; however, only some abundant amino acids such as Pro, Arg, Asn, Gln and GABA are involved in stress tolerance mechanisms as compatible osmolytes, precursors of secondary metabolites, or storage forms of organic nitrogen. On the contrary, low abundant amino acids accumulate under stressful conditions mostly due to protein degradation [42].

This work aimed at a better understanding of the mechanisms of salt tolerance in halophytes of the genus *Limonium*. Apart from increasing our basic knowledge on these mechanisms, this study is relevant also for conservation purposes, as the information obtained could be useful for the conservation and reintroduction programmes of these two endangered species. The specific objectives were to check the ability of seeds of the two species to germinate under saline conditions and to maintain their germination capacity after a period of exposure to salt, as well as to compare their limits of salt tolerance during vegetative growth. To address the latter question, growth parameters, chlorophyll degradation and the pattern of variation of different mono and divalent ions were determined in plants of the two species subjected in the greenhouse to salinity levels similar and beyond those existent in their natural environments. As the two species showed different degrees of tolerance to salinity, we hypothesised that this would be reflected in qualitative and quantitative differences in their metabolic profiles. Therefore, the work was completed with an untargeted metabolomics study of primary metabolites (sugars and polyols, phosphoric and organic acids, and amino acids) in control and stressed plants. The analysis of salt-induced changes in the metabolite profiles can provide relevant information on the possible mechanisms of salt tolerance in the investigated *Limonium* species.

## 2. Results

### 2.1. Seed Germination

Final germination percentages after 30 days of incubation were very high in the controls (seeds germinated in water) for the two species, 95% in *L. albuferae* and 98% in *L. dufourii*. Salinity inhibited seed germination in both species, reducing the germination percentages to ca. 30% in the presence of 150 mM NaCl. A significant difference between the two species was found at 300 mM NaCl, with *L. albuferae* showing a higher percentage of germinated seeds (18.5%) than *L. dufourii* (2.5%). Only a few seeds of the two species germinated in the presence of 450 mM NaCl, whereas 600 mM NaCl completely inhibited germination (Figure 1). Most seeds that did not germinate after one month in the presence of salt in the initial germination assays, recovered their germination capacity when they were washed with water and transferred to new Petri dishes. Germination percentages after one additional month of incubation in water were elevated, ranging from 60% to 80% in *L. albuferae* (Figure 1a) and from 80% to ca. 95% in *L. dufourii* (Figure 1b). The lower values corresponded to seeds that had been pre-incubated with 150 mM NaCl, whereas the higher germination percentages were measured for those recovered from the 300, 450, and 600 mM NaCl plates, without statistically significant differences between these pre-treatments (Figure 1).

As expected, germination velocity decreased with increasing salinity, as shown by the parallel increase in the mean germination time (MGT, Table 1); the seeds of *L. dufourii* generally germinated faster than those of *L. albuferae*, but the pattern of concentration-dependent variations was more irregular. Germination in water in the ‘recovery’ assays was generally faster than in the initial germination experiments, with small differences in the calculated MGT values, irrespective of the salt concentration used in the seed pre-treatments (Table 1).

### 2.2. Growth Performance under Salt Stress Conditions

Salt stress inhibited growth in both *Limonium* species (Figure 2), affecting mostly the aerial part of the plants, as indicated by a significant reduction in the number of leaves and the leaf fresh weight, in relation to the corresponding controls (Table 2). Comparing the two species, *L. dufourii* showed a more accentuated reduction of these growth parameters in parallel with increasing external salinity. For example, leaf fresh weight (FW) of *L. albuferae* plants treated with 200 mM NaCl did not differ significantly from the non-stressed controls, whereas the same concentration caused a reduction of about 50% in *L. dufourii*.

The first species tolerated better all salt concentrations, and a drastic inhibitory effect was noticed only in the presence of 600 mM NaCl, when plants lost half of their aerial FW (Table 2). The reduction of leaf FW was partially due to loss of water, as the salt treatments induced a slight (but significant) dehydration of the plants; in this case, however, very small differences were observed between the two species (Table 2). Interestingly, root FW increased significantly, especially in *L. dufourii*, in the presence of moderate (200 mM) or relatively high (400 mM) NaCl concentrations, to decrease again at even higher salinities (600–800 mM NaCl), but never below the values measured in the control plants (Table 2). Moreover, no salt-induced dehydration was detected in roots, as their water content did not suffer any significant reduction in either species.

Regarding photosynthetic pigments, salt stress induced a significant, concentration-dependent reduction in the levels of chlorophylls a (Chl a) and b (Chl b), and carotenoids (Caro) in *L. dufourii*. On the contrary, in *L. albuferae* Chl a contents remained practically unchanged at all NaCl concentrations tested, as the small variations with respect to the controls were not statistically significant; Chl b and Caro contents did show a reduction when comparing control and salt-stressed plants, but without significant differences between the values obtained at different salinities (Table 2). These findings are in agreement with the responses to salt of the two species in terms of growth, indicating a relative higher tolerance to salinity in *L. albuferae*.

### 2.3. Ion Accumulation

Plant Na^+^ and Cl^−^ contents increased in parallel to increasing external salinity in the two *Limonium* species, both in roots and leaves. In general, at all external NaCl concentrations tested, the levels of these two ions were higher in leaves than in roots, and in *L. dufourii* than in *L. albuferae*, except for root Cl^−^ contents, which were similar in both species. Also, in most cases, Na^+^ accumulated to higher levels than Cl^−^, for each particular species, organ and salt treatment (Table 3). Regarding K^+^ levels, they were also higher in leaves than in roots in the two species, and did not vary significantly in response to salt stress in roots of *L. albuferae*; the pattern of salt-induced changes in K^+^ contents was different in *L. dufourii* roots, where they first decreased at 200 and 400 mM NaCl, to increase again at higher salinities, reaching control values in the presence of 800 mM NaCl. When leaf K^+^ contents are considered, however, the two *Limonium* species responded in the same way to salt stress, with a significant reduction with respect to the controls, but no differences between the different salt treatments applied (Table 3). Finally, bioavailable Ca^2+^ concentrations were also measured in the samples, showing salt-induced patterns of variation qualitatively similar to those of Na^+^ and Cl^–^: higher Ca^2+^ contents in leaves than in roots, and a concentration-dependent increase in the levels of this cation in response to increasing salinity, in roots and leaves of both species (Table 3). It is important to highlight the relatively high concentrations of all measured ions in the leaves of control, non-stressed plants, generally much higher than the corresponding values in roots (Table 3).

### 2.4. Statistical Analysis of the Differences in Growth Parameters, Photosynthetic Pigments and Ion Accumulation between the Two Species

A factorial ANOVA was performed, considering three sources of variation: Species (A), Treatment (B) and Organ (C). The effects of the three factors and their interactions are summarised in Table 4. This analysis revealed that the most distinctive trait for assessing the effect of salinity in the two species was FW, which varied significantly according to the three parameters mentioned above: the salt concentration applied, the organ (roots or leaves), and also on a genetic basis, that is, according to species. Therefore, we consider FW as a reliable morphological marker of salt tolerance in the selected *Limonium* species.

The interactions between the effects of ‘species’ and ‘treatment’, and between ’organ’ (roots or leaves) and ‘treatment’, are shown in Figure 3. Both species exhibited different responses as salinity increased. *L. albuferae* was more tolerant than *L. dufourii* to NaCl concentrations between 200 and 400 mM (Figure 3a), showing higher FW values (Figure 3a). However, at salt concentrations of 600 or 800 mM NaCl both species were equally sensitive (Figure 3a). Comparing the FW of leaves and roots (Figure 3b), a different pattern could be observed. Root FW slightly increased at low salinity (between 0 and 200 mM NaCl) in response to the salt treatment, whereas a progressive reduction of FW with increasing salinity was observed in leaves. The pattern of WC% variations supported the above results: up to 600 mM external NaCl, a stronger dehydration was observed for *L. dufourii* as compared to *L. albuferae*, which showed a more progressive reduction of WC (Figure 3c). A concentration-dependent loss of water was detected in plant leaves in response to increasing external salinity, whereas the pattern of salt-dependent WC reduction was more irregular in roots (Figure 3d).

Regarding the photosynthetic pigments, the effect of species was significant for Chl b and Caro, but not for Chl a, whereas the effect of the treatment was significant for Chl a, and the interaction species x treatment only for carotenoids (Table 4).

Ion contents were strongly influenced by the factor ‘species’ and, with the exception of K^+^, also by the organ, roots or leaves. As expected, the effect of the salt treatment was highly significant for the accumulation of Na^+^ and Cl^−^, as were many of the interactions between different factors (Table 4).

### 2.5. Differences in the Metabolite Profiles of the Two Species in the Absence of Stress

The metabolite profiles were analysed by Gas Chromatography-Mass Spectrometry in control plants, not subjected to the salt stress treatments. Three categories of primary metabolites were detected: carbohydrates, organic acids (together with phosphoric acid) and amino acids (Figure 4). Seven carbohydrates were found in the two species (glycerol, rhamnose, fructose, glucose, *myo*-inositol, raffinose and sucrose), and one (erythritol) was detected only in *L. albuferae*. The relative contents of most carbohydrates were roughly similar in the two *Limonium* species, except for raffinose, which was much more abundant in *L. albuferae* (Figure 4a), and for fructose and glucose, present at higher relative levels in *L. dufourii* (Figure 4b).

Phosphoric acid and six organic acids (glyceric, succinic, maleic, malic, threonic and citric) were detected in the two species, without marked differences observed in their relative contents in both of them (Figure 4c,d).

Of the 17 amino acids detected, 12 were found in the two species: Ala, Val, Leu, Ile, Gly, Ser, Thr, GABA, Asp, Asn, Trp and Pro. Pyro-Glu, Glu and Gln were detected only in *L. albuferae* (Figure 4e) and Lys and Phe only in *L. dufourii* (Figure 4f). Significant quantitative differences between the two species were observed regarding the relative contents of most amino acids. The most abundant amino acid in *L. albuferae*, showing substantial differences with all the others, was Glu—not detected in *L. dufourii*—whereas very low levels of Trp and Pro were measured in this species (Figure 4e). In *L. dufourii* there was no single predominant amino acid, as four of them (Ala, Gly, GABA and Asp) showed similarly high relative contents and, in all four cases, significantly higher than in *L. albuferae.* Very low Trp and Pro relative contents were also measured in *L. dufourii* (Figure 4f).

### 2.6. Changes in Metabolites Relative Contents in Response to Salt Stress

The salt treatments induced changes in the relative contents of many of the metabolites mentioned above, with quantitative and qualitative differences often observed between the two *Limonium* species. A summary of all results is presented in Appendix A. Concerning sugar and polyol relative contents, in both *Limonium* species erythritol, rhamnose and sucrose increased with increasing salt concentrations, glycerol showed oscillations, generally not significant, and *myo*-inositol levels decreased in response to salt stress; glucose and fructose relative levels were higher in non-stressed *L. dufourii* plants, but showed significant increases over control values only in *L. albuferae*.

The relative levels of phosphoric acid, and the organic acids glyceric and threonic increased in salt-stressed plants of the two species. In the case of threonic acid, when comparing the two species, the most statistically significant increase was observed in *L. dufourii* plants treated with 400 mM NaCl. No significant changes in the foliar contents of maleic, malic and citric acids were detected in *L. albuferae* in response to increasing external salinity, whereas a general decrease was observed in *L. dufourii* regarding the relative contents of these compounds. Succinic acid showed an irregular oscillation pattern in both species.

A general trend of increasing amino acid relative contents under salt stress was detected for most amino acids in both species, but especially in *L. albuferae*: Ala, GABA, Gly, Ile, Leu, Lys, Phe, Pro, Ser, Thr, Trp and Val. For example, in *L. albuferae*, Phe and Trp increased from undetected levels in the control plants to maximum values in the presence of 800 mM NaCl; in *L. dufourii*, the same amino acids showed a peak at 400 mM NaCl, decreasing at higher salinities. The same accumulation pattern—i.e., maximum contents at the highest salt concentrations tested (600–800 mM NaCl) in *L. albuferae* and a peak at 400 mM in *L. dufourii*—was observed for other amino acids, such as Asn, Asp, Gly, Ser, Thr or Val. In fact, a statistically significant increase was observed for these amino acids along with Ile, Leu, Phe and Pro in *L. albuferae* plants upon maximum salt stress (600–800 mM NaCl), when comparing both species. Besides, Glu, Gln and pyro-Glu were only found in *L. albuferae*, but their levels did not vary significantly, or the changes were not clearly correlated with the external salt concentration.

### 2.7. Statistical Analysis of the Untargeted Metabolomics Results

A two-way ANOVA was performed to analyse the effects of the treatment, species and their interactions, for each of the identified metabolites (Table 5). Concerning the carbohydrates group (sugars and polyols), the effect of the ‘species’ factor was significant for fructose, erythritol, rhamnose and sucrose, and that of ‘treatment’, for all detected carbohydrates except glycerol. The highest interactions of the two factors were observed for raffinose and erythritol, which varied in different directions (increasing/decreasing with salinity) in the two species.

The two-way ANOVA also revealed a significant effect of ‘species’ for two organic acids, citric and threonic, and that of ‘treatment’ for glyceric and threonic acids (Table 5). Regarding the amino acids, the ‘species’ effect was highly significant for Glu, Gln and pyro-Glu—as expected since these amino acids were only detected in *L. albuferae*—but also for Ile, Lys, Thr and Val. The effect of the treatment was significant for all detected amino acids except GABA, Glu and Gly (Table 5).

A comparison of all samples included in the metabolomics analysis was performed through multivariate data analysis (MVDA). Specifically, a partial least square (PLS) analysis was applied, defining as the X variable the area of the characteristic ion mass spectra, and as the step-wise Y variables the species (*L. albuferae* and *L. dufourii*) and the treatments (control and salt stress). The score plot of PLS analysis according to model terms clearly separated the effect of treatment in the OX axis (explaining 29.5% of the total variability) and that of species in the OY axis (11% of variability). When analysing the distribution of plants from different salt treatments (Figure 5), control and 200 (200 mM NaCl) of the two species were grouped together with 400 of *L. albuferae* in the negative part of PLS component 1, whereas 400 in *L. dufourii* falls together with higher concentrations (600 and 800) of *L. albuferae* and were placed in the positive side. The second component of PLS clearly separated the metabolite relative contents of both species, locating *L. dufourii* (LD) and *L. albuferae* (LA) in the positive and in the negative part, respectively. This analysis indicated that the salt stress provoked important changes in the metabolic profiles of both species, being different in each one.

The loading scatter plot of PLS analysis (Figure 6) showed that Pro, rhamnose, erythritol, threonic acid and sucrose were the metabolites that mostly increase in conditions of salinity whereas Thr, citric, malic and maleic acids are those most related to the species. Out of them, Thr was the only metabolite that showed a peak at lower salt concentration in *L. albuferae* (at 200 mM NaCl) than in *L. dufourii* (400 mM NaCl), and the remaining ones showed also a different pattern of variation in the two species.

## 3. Discussion

### 3.1. Relative Salt Tolerance of Limonium Albuferae and L. Dufourii

As highlighted in the introduction, the two species under study present a special interest, as they are unique endemics from salt marshes in the area of Valencia. Such ecosystems are highly dynamic, characterised by large variations in the salinity of the soil, at the temporal and spatial scales [43]. Therefore, reintroduction or reinforcement of populations should be based on a deep knowledge not only of the soil characteristics and their seasonal variations, but also on the tolerance limits of the species of interest. Increased temperature due to global warming leads to increased evapotranspiration, which can generate hypersaline conditions; even acting over a short time period, this may inflict a greater stress of plants living in salt marshes, eventually causing, in the worst-case scenario, the dieback of those less tolerant [44] The two species of *Limonium* studied in the present work share the same area in l’Albufera Natural Park, the site of origin of their seeds, but they differ in their tolerance to salt stress. *Limonium albuferae* proved to be more salt tolerant than *L. dufourii*, as a higher percentage of its seeds germinated at 300 mM NaCl, its growth was not affected in the presence of 200 mM NaCl, and higher salt concentrations (400 and 600 mM) inhibited vegetative growth to a lesser extent than in *L. dufourii*. Seed germination and seedling establishment are critical phases in the life cycle of halophytes [45] and a reduction of the salinity in the superficial soil layers, to allow germination, is a prerequisite for their survival in salt marshes. The concentration of 200 mM NaCl drastically affects or, in combination with high temperatures, is even the salinity limit allowing germination for several Iberian *Limonium* endemics [46,47,48]. On the contrary, in other species of this genus seeds still maintain their germination ability at higher salt concentrations [49,50,51], as in other halophytes [52,53,54,55]. However, a distinctive trait of halophtyes is the recovery of germination capacity even after long periods of exposure to high salt concentrations, a phenomenon that does not occur in glycophytes [56]. The cessation of dormancy triggered by the alleviation of salt stress is significant in salt marshes, which are subjected to seasonal fluctuations of salinity, [57,58]. Although *L. dufourii* showed lower germination rates under saline conditions, after the salt pre-treatments its seeds had an excellent recovery capacity, reaching germination percentages in water as high as in the control and up to ~95%, whereas the percentage of recovery ranged from 60% to 80% in *L. albuferae*. The ecological significance is that seeds of both species maintain their viability under high soil salinity conditions and at this stage their responses to salt stress are similar to those of other *Limonium* species.

During vegetative growth, once the critical phase of seedling establishment is overcome, the ability to tolerate different types of stresses, including salinity, increases in parallel to the age of plants [52]. For this reason, an optimal method to assess the limits of salt tolerance of one species, but also to compare stress tolerance of genetically related taxa, is the quantification of the salt-induced inhibition of growth of the plants [59,60]. Growth reduction under salt stress is a general trait in glycophytes, but also in most halophytes. However, in many dicotyledonous halophytes, especially in those more salt-tolerant, moderate concentrations of NaCl stimulate growth [61]. Stimulation of growth under moderate salinity conditions has also been reported in some *Limonium* species, such as *L. delicatulum* [62], *L. girardianum* or *L. virgatum* [49]. Of the two species studied here, only *L. albuferae* experienced an increase of FW in the presence of 200 mM NaCl, whereas plants of *L. dufourii* subjected to the same treatment suffered a reduction of growth of more than 50%. In fact, all analysed growth parameters clearly indicated that *L. dufourii* is more sensitive to salt during vegetative growth than *L. albuferae*; this finding should be considered in the management programmes of these two endangered species. Plants of both species appear to be tolerant to soil salinities much higher than those normally present in the salt marshes that represent their natural habitats, considering that they survived for one month in the greenhouse in the presence of NaCl concentrations as high as 600–800 mM. This explains how these plants can withstand in the field episodes of hypersalinity, occurring generally in summer due to increased temperatures and evapotranspiration, as it has been reported for the area of study [41,63,64].

### 3.2. Main Salt Tolerance Mechanisms in the Investigated Species

The strategy of using inorganic ions as cheap osmotica to maintain growth under saline conditions, is a relevant mechanism of salt tolerance in many halophytes, including *Limonium* species [49,65,66,67,68]. In our experiments, ion contents were generally higher in leaves than in roots, at all external salinities tested, in agreement with those previous data. Accumulation of Na^+^ in plants is generally associated with a decrease of K^+^ levels, as it has been found in the present work in the leaves of the two *Limonium* species. The drop in K^+^ levels is mainly due to the competition of Na^+^ for the same binding sites in proteins, including the physiological K^+^ membrane transport proteins. Moreover, increased Na^+^ concentrations induce depolarisation of the plasma membrane, causing the loss of cellular K^+^ by activation of outward rectifying K^+^ channels [69]. In the selected *Limonium* species, K^+^ levels were constant in roots but progressively decreased in leaves, in parallel with increasing external salinity. The reduction in K^+^ was more pronounced in *L. dufourii*, which also showed a relatively higher accumulation of Na^+^ in salt-stressed plants, in comparison to the controls. For this reason, the increase in the leaf Na^+^/K^+^ ratio in plants grown in the presence of 800 mM NaCl (the highest salinity tested) with respect to the controls, was much higher in *L. dufourii* (~27-fold) than in *L. albuferae* (~2.5-fold). Maintaining a balanced cytosolic Na^+^/K^+^ ratio is regarded as an essential mechanism for salt tolerance [70], and it seems logical to assume that the lower Na^+^/K^+^ ratios in *L. albuferae* contribute to its better tolerance to high salinity as compared to *L. dufourii.*

The observed increase in Ca^2+^ contents in response to increasing external salinity, both in roots and in leaves of the two *Limonium* species is probably also involved in the salt tolerance mechanisms of these species, as calcium plays a key regulatory and signalling role in plant growth and development under salt stress [71]; for instance, extracellular Ca^2+^ is beneficial for maintaining Na^+^ and K^+^ homeostasis via the SOS pathway [72]. Moreover, an increase in Ca^2+^ concentration stimulated the development and the salt-secretion rates of salt glands in the leaves of *L. bicolor*; salt secretion is an effective strategy used by recretohalophytes (as the two species studied here) to adapt to highly saline soils [73].

It is interesting to mention the relatively high concentrations of all measured ions in leaves of control, non-stressed plants, generally much higher than the corresponding values in roots. These data support the existence of constitutive mechanisms of tolerance based in the active transport of inorganic ions to the leaves to contribute to osmotic adjustment, even under low soil salinity conditions.

The primary metabolites that were detected in the untargeted metbolomics analysis belong to the groups of carbohydrates (sugars and polyols), organic acids, together with phosphoric acid, and amino acids. Carbohydrate profiles were qualitatively similar in the two analysed species, in the absence of stress. Several of the identified compounds did not vary significantly, or did not show a clear correlation of the variation patterns with the intensity of the salt treatments, whereas the leaf relative contents of erithritol, sucrose and rhamnose clearly increased in parallel to increasing external salinities in the two species. Fructose, which has been reported as a major osmolyte in different *Limonium* species [49,74,75], showed a salt-induced increment only in *L. albuferae*, and a similar pattern was observed for glucose. Therefore, the relatively higher salt tolerance of *L. albuferae* may be partly due to the specific accumulation of fructose and glucose under high salinity conditions in this species, but not in *L. dufourii.*

Among the organic acids identified in the leaf extracts, the most abundant in the two species were malic, citric and maleic. Their concentrations did not vary significantly when *L. albuferae* plants were subjected to the salt treatments, but they decreased significantly in *L. dufourii*. Citric and malic acids also decreased in salt-treated *L. latifolium* plants [74]. On the contrary, citric acid levels increased upon salt stress in the halophyte *Leymus chinensis*, and improved growth when applied exogenously. Higher constitutive levels of citric and malic acids were detected in the salt-tolerant *Thelungiella halophilla* in comparison to *Arabidospis thaliana* [76]. These and other data support the positive role of citric and malic acids in the mechanisms of salt tolerance and, accordingly, it can be suggested that *L. albuferae* respond to salt stress better than *L. dufourii*, in part, by maintaining steady levels of these two organic acids.

Amino acids are synthesised by various distinct metabolic pathways, some strongly influenced by environmental conditions. For example, the glutamate family pathway is strongly activated under stress, leading to the accumulation of Pro and GABA [77]. Pro is the commonest osmolyte in plants, directly participating in cellular osmotic adjustment under stress conditions, but also playing additional roles as ‘osmoprotectant’—low-molecular-weight chaperon and ROS scavenger [78]. Together with fructose, Pro has been reported as the main osmolyte in four species of *Limonium* from the area under study [49], but it seems to play only a minor osmoregulatory role in *L. latifolium* [74]. Besides its function in maintaining carbon-nitrogen balance, GABA is involved in responses to different types of abiotic stress in plants. Increased concentrations of GABA were detected in conditions of hypoxia, drought, salinity and low or high temperatures, and its role as effective osmolyte and in ROS scavenging is well-known [79].

The metabolite profile of *L. albuferae* and *L. dufourii* samples showed, as a general trend, a salt-induced increase in the levels of most amino acids, especially in the former species, suggesting a positive role of these compounds in the mechanisms of salt tolerance of both *Limonium* species. GABA and Pro contents gradually increased in the two species in parallel with the external salt concentration, reaching maximum values in the presence of 800 mM NaCl. A similar pattern, with peaks at 800 mM or 600 mM NaCl, was observed for many other amino acids in *L. albuferae*, whereas in *L. dufourii* maximum concentrations were measured in the 400 mM NaCl treatment, decreasing at higher salinities. This pattern of variation correlates with the relative tolerance of the two species: It appears that, at high salinities, the more salt-sensitive *L. dufourii* cannot use as efficiently as *L. albuferae* this mechanism of defence based on the accumulation of specific amino acids.

### 3.3. Relevance of the Obtained Results for Conservation Strategies of the Two Endemic Limonium Species

As stated in the Introduction, *L. albuferae* and *L. dufourii* are highly threatened local endemics, represented by a few populations with a rather low number of individuals in salt marshes located near Valencia. Both species require management programmes for ensuring their persistence, mostly due to habitat loss but also to other factors, including for example pressure of invasive plants, which should be taken into consideration when establishing new populations in the frame of reintroduction strategies. The data presented here indicate that soil salinity, per se, is not a restrictive factor as both species are halophytic, although with different degrees of salt tolerance. The more tolerant *L. albuferae* could be reintroduced in the most saline areas where there will be less competition with aggressive invasive species, such as those of the genus *Spartina***,** which are increasing in the area (personal observations). On the other hand, new sites for *L. dufourii* should be considered only in areas with moderate salinity, as its salt tolerance is lower than in other species of the genus [49]. Given the uncertainty of climate change effects, conservation efforts of the two species—but especially of the less tolerant *L. dufourii*—should include continuous monitoring of soil salinity and moisture in the areas of location or reintroduction, as well as the presence and evolution of invasive plants.

## 4. Materials and Methods

### 4.1. Seed Germination

Seeds of *L. albuferae* and *L. dufourii* were provided by Centro para la Investigación y Experimentación Forestal (Centre for Forest Research and Experimentation, CIEF). Before germination, seeds were disinfected with a 10% hypochlorite solution and then thoroughly washed with distilled water. For the germination assays, four replicas of 20 seeds were placed in standard Petri dishes (90 mm in diameter) with a sterile cotton layer covered by two layers of filter paper, which had been wetted with 20 mL of distilled water (for the control treatments) or NaCl solutions of increasing concentration (150, 300, 450 and 600 mM). The plates were incubated in a germination chamber (Equitec, EGCHS HR), with a photoperiod of 16 h of light at 30 °C, and 8 h of darkness at 20 °C. The number of germinated seeds was counted every two days, considering a seed as germinated when the radicle emerged and reached 2–3 mm. After 30 days, all seeds that did not germinate were rinsed several times in distilled water and transferred to new Petri dishes for the ‘recovery of germination’ assays in distilled water. The number of germinated seeds was recorded for another 30 days. The percentages of germination were calculated as means of the four replicates for each species and treatment. The germination velocity was estimated by calculating the ‘Mean Germination Time’ (MGT), as described by Ellis and Roberts [80]:

MGT= Σ D *n*/Σn, where D is ‘days from the beginning of the germination test’, and n is the number of seeds newly germinated on day D.

### 4.2. Plant Growth and Salt Treatments

Seeds without any prior sterilisation were sown on a mixture of commercial peat and vermiculite (3:1) and watered occasionally with Hoagland nutrient solution [81]. After three weeks, seedlings were transferred to individual 1 L pots placed in plastic trays, with five pots per tray. One week later, salt treatments were started, by watering the plants with water (for the control treatments) or with aqueous salt solutions containing NaCl at 200-, 400-, 600- and 800-mM final concentrations (for the salt stress treatments). In all cases, 1 L of water or the corresponding salt solution was added to each tray every five days. Five biological replicas (five individual plants) were used per species and per treatment. All experiments were conducted in a controlled environment chamber in the greenhouse, under the following conditions: long-day photoperiod (16 h of light), temperature of 23 °C during the day and 17 °C at night, and 50–80% relative humidity.

After one month of treatment, the aerial parts and the roots of the plants were harvested and weighed separately. The following plant growth parameters were measured: Fresh weight of leaves (FWL) and roots (FWR), water content percentage of leaves (WCL) and roots (WCR), and leaf number (LN). Water content percentage in leaves was calculated as: WC% = [(FW-DW)/FW] × 100. For an easier comparison of the two species, which differ in size, fresh weights of leaves and roots were expressed in percentages of the corresponding mean values of the control, non-stressed plants, considered as 100%.

### 4.3. Photosynthetic Pigments

Chlorophylls a and b (Chl a, and Chl b) and total carotenoids (Caro) contents, were determined as described by Lichtenthaler and Wellburn [82]. Pigments were extracted from 0.05 gr of fresh leaf material in 10 mL of ice-cold acetone 80%. After mixing overnight and centrifuging for 10 min at 12,000 rpm, the supernatant was collected, and its absorbance was measured at 663, 646 and 470 nm. Pigment concentrations were calculated using the equations formulated by Lichtenthaler and Wellburn [82], and their contents were expressed in mg g^−1^ DW.

### 4.4. Ion Content Measurements

Ion contents were determined in roots and leaves, after being eluted in aqueous extracts according to the protocol of Weimberg [83], by heating the samples (0.05 g of dried, ground plant material in 15 mL of water) for 15 min at 95 °C in a water bath, followed by filtration through a 0.45 µm filter (Gelman Laboratory, PALL Corporation). Cations Na^+^ and K^+^ and bioavailable Ca^2+^ were quantified with a PFP7 flame photometer (Jenway Inc., Burlington, VT, USA), Cl^–^ was measured using a chloride analyser Corning 926.

### 4.5. Primary Metabolite Extraction and Metabolite Profiling Analysis

Primary metabolite analysis was performed by the Metabolomics Platform of the “Institute for Plant Molecular and Cell Biology” (Polytechnic University of Valencia, Spain) as previously described [84] with some modifications, following the platform’s standard procedures. Each sample (100 mg of leaf material) was homogenised in liquid N_2_ and extracted with 1.4 mL 100% methanol and 60 µL internal standard (0.2 mg ribitol in 1 mL of water). The mixture was extracted for 15 min at 70 °C; subsequently, the extract was centrifuged for 10 min at 14,000 rpm. The supernatant was transferred to a glass vial, and 750 µl of chloroform and 1.5 mL of water were added. The mixture was vortexed for 15 s and centrifuged for 15 min at 14,000 rpm. Finally, 150 µL aliquots of the methanol/water upper phase were dried in vacuo for 6–16 h.

For derivatisation, dry residues were redissolved in 40 µL of 20 mg/mL methoxyamine hydrochloride in pyridine and incubated for 90 min at 37 °C, followed by addition of 70 µL MSTFA (N-methyl-N-[trimethylsilyl]trifluoroacetamide) and 6 µL of a retention time standard mixture (3.7% [*w*/*v*] mix of fatty acid methyl esters ranging from 8 to 24 C) and further incubation for 30 min at 37 °C. Sample volumes of 2 µL were injected in split and splitless mode, to increase metabolite detection range, in a 6890 N gas chromatograph (Agilent Technologies Inc. Santa Clara, CA) coupled to a Pegasus 4D TOF mass spectrometer (LECO, St. Joseph, MI). Gas chromatography was performed on a BPX35 (30 m × 0.32 mm × 0.25 μm) column (SGE Analytical Science Pty Ltd., Australia) with helium as carrier gas, at a constant flow of 2 mL/min. The liner was set at 230°C. The oven programme was 85 °C for 2 min, followed by an 8 °C/min ramp up to 360 °C. Mass spectra were collected at 6.25 spectra s^−1^ in the *m*/*z* range 35–900 and ionisation energy of 70 eV. Chromatograms and mass spectra were evaluated using the CHROMATOF programme (LECO, St. Joseph, MI).

### 4.6. Statistical Analysis

Percentages of germination were arcsine transformed prior to the statistical analysis. Analysis of variance was carried out separately on data from each species. When the ANOVA null hypothesis was rejected, differences between treatments were evaluated by the Tukey’s test.

A multifactorial ANOVA was applied to assess the interaction between species, treatments and organs (leaves or roots). Data were analysed using the software Statgraphics Centurion v.16 (Statpoint Technologies, Warrenton, VA, USA).

For the untargeted NMR-metabolomics analysis, the 1H-NMR spectra were automatically reduced by AMIX (v. 3.7, Bruker Biospin) to ASCII files. Partial least square (PLS), and orthogonal projection to latent structures-discriminant analysis (OPLS-DA) were performed with the SIMCA-P software (v. 13.0.3, Umetrics, Umeå, Sweden) using the Pareto scaling method. For the targeted GC-MS-metabolomics analysis, the area of each primary metabolite relative to the ribitol area was used as X variable of Principal Component Analysis (PCA) using unit variance (UV) scaling method.

## 5. Conclusions

The germination study indicated that both species behave as halophytes, their germination being enhanced after one month of exposure to salt concentrations. *Limonium albuferae* proved to be more salt tolerant than *L. dufourii*, with higher germination percentage at 300 mM NaCl, but higher salt concentrations completely inhibited germination in both species. The same pattern was observed during vegetative growth, as *L dufourii* was more affected by salt treatments than *L. albuferae*, which had an optimal growth when watered with 200 mM NaCl. Other parameters, such as photosynthetic pigments and the Na^+^/K^+^ ratio also indicated that *L. albuferae* can better tolerate saline conditions. The metabolomics analysis revealed only a few differences between the two species in absence of stress, in plants from the control treatment, but the pattern of accumulation of several compounds under stress was different. Fructose and glucose, the first reported as a major osmolyte in the genus, were accumulated only in *L. albuferae*, which also showed steady levels of citric and malic acids. Proline and γ-aminobutyric acid (GABA), both with osmoregulatory functions, increased in both species.

## Figures and Tables

**Figure 1 metabolites-09-00294-f001:**
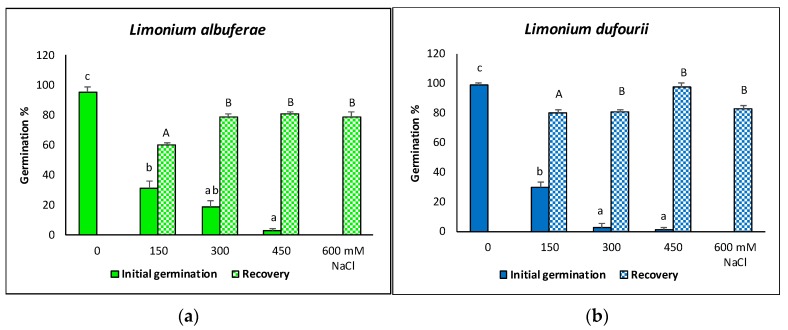
Final germination percentages in the two *Limonium* species after 30 days exposure to the NaCl concentrations indicated in the graphs: (**a**) *L. albuferae,* (**b***) L. dufourii*. Bars represent mean ± SE values (*n* = 4). Same letters indicate homogeneous groups between treatments for each species (*p* < 0.05). Lower-case letters were used for initial germination in the presence of salt and capital letters for the recovery of germination.

**Figure 2 metabolites-09-00294-f002:**
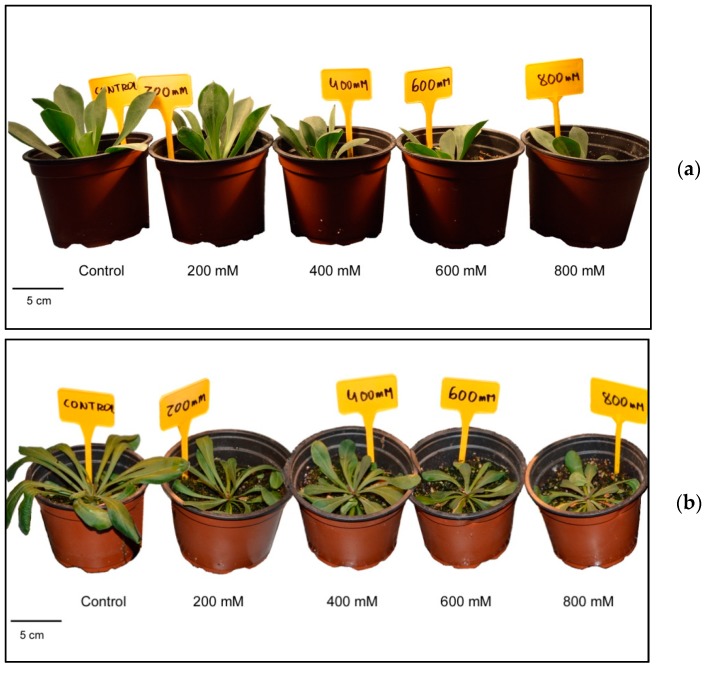
Effects of 30 days salt treatments on growth of young plants of *L. albuferae* (**a**) and *L. dufourii* (**b**). Treatments started one month after sowing the seeds.

**Figure 3 metabolites-09-00294-f003:**
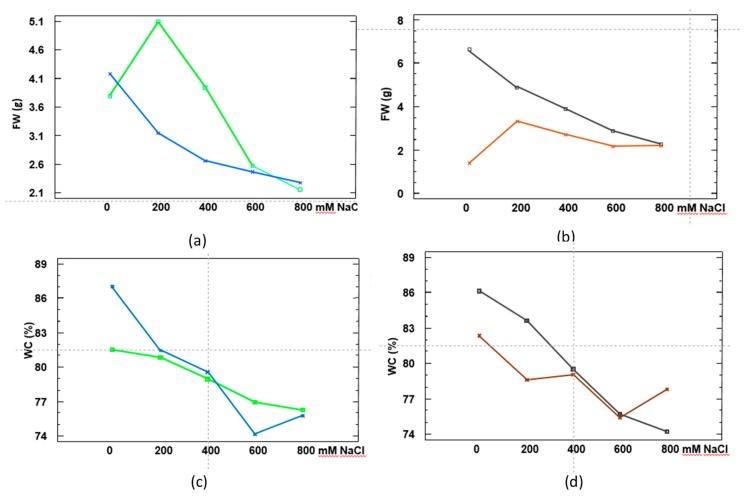
Three-way interaction plots (Species × Treatment × Organ) for fresh weight (FW) (**a**,**b**) and for water content (WC) (**c**,**d**) in plants of *L. albuferae* (green), *L. dufourii* (blue), leaves (black) and roots (brown). In **a** and **b**, interactions Species x Treatment are shown (not differentiating ‘organs’, leaves and roots, values); in **b** and **d**, interactions Organ x Treatment are shown (not differentiating ‘species’ values).

**Figure 4 metabolites-09-00294-f004:**
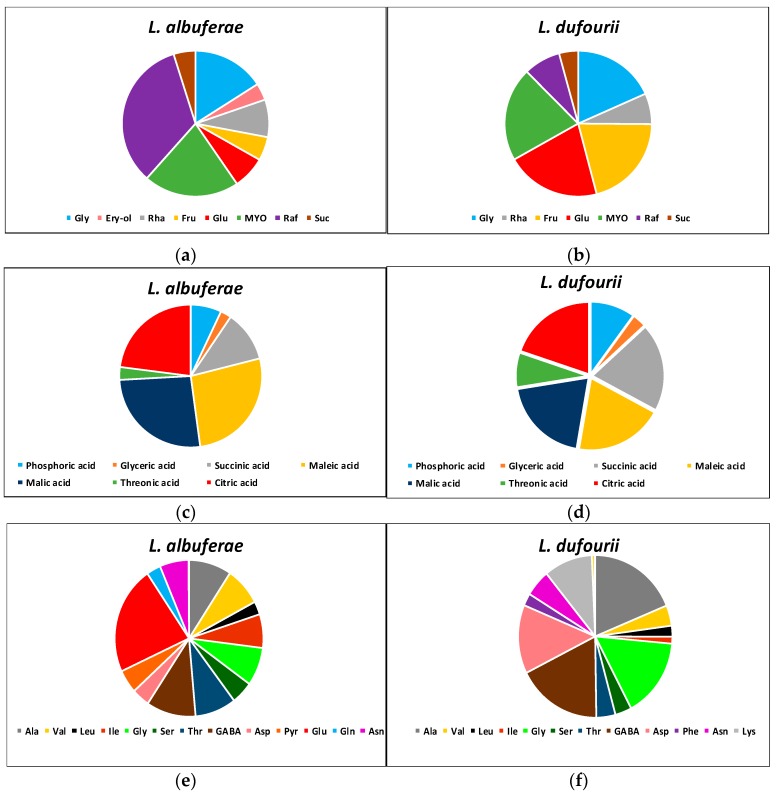
Relative contents of leaf carbohydrates (**a**,**b**), phosphoric and organic acids (**c**,**d**) and amino acids (**e**,**f**) detected in absence of stress in plants from control treatments (*n* = 5) of the two *Limonium* species.

**Figure 5 metabolites-09-00294-f005:**
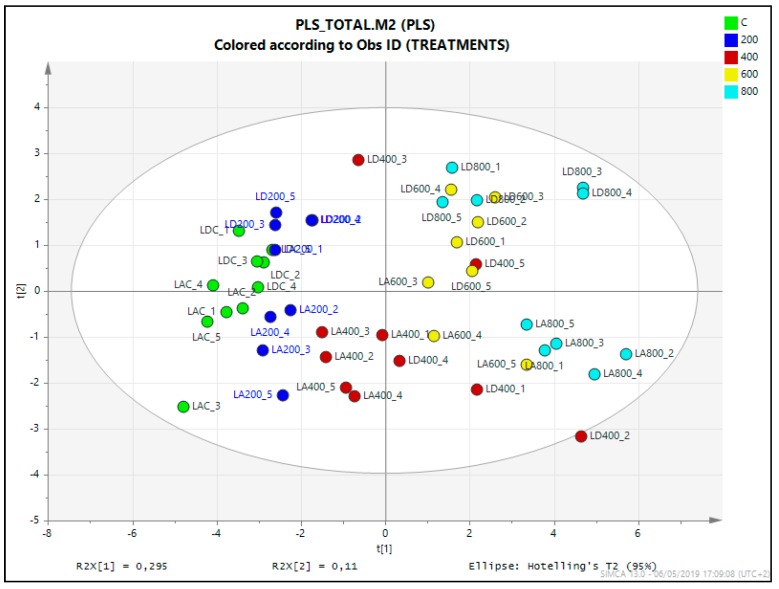
Score plot of partial least square analysis (PLS) based on the characteristic ion of the mass spectra from the primary metabolites measured in the m/z range 35–900, of the *L. albuferae* (LA) and *L. dufourii* (LD) control (C) plants and after salt stress (200, 400, 600, and 800 mM NaCl).

**Figure 6 metabolites-09-00294-f006:**
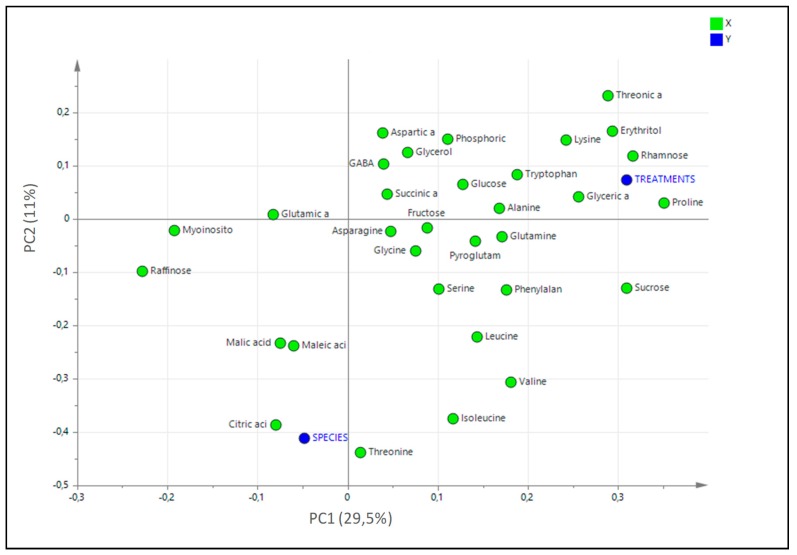
Loading scatter plot analysis of PLS analysis of the showing the metabolites involved in salt stress responses in the metabolomics study in the two *Limonium* species.

**Table 1 metabolites-09-00294-t001:** The seed mean germination time (MGT) (in days, in the initial germination and recovery of germination assays). Mean ± SE values are shown (*n* = 4). Same letters indicate homogeneous groups between treatments for each species according to the Tukey test (*p* < 0.05).

NaCl (mM)	*Limonium albuferae*		*Limonium dufourii*	
	Initial Germination	Recovery	Initial Germination	Recovery
0	5.78 ± 0.33 ^a^	-	5.32 ± 0.38 ^a^	-
150	10.86 ± 0.78 ^b^	4.41 ± 0.15 ^ab^	11.99 ± 1.16 ^bc^	5.97 ± 0.45 ^b^
300	16.67 ± 0.59 ^c^	4.41 ± 0.20 ^ab^	6.50 ± 0.71 ^ab^	4.88 ± 042 ^ab^
450	20.00 ± 1.41 ^c^	4.51 ± 0.17 ^b^	13.00 ^c^	3.89 ± 0. 28 ^a^
600	-	3.73 ± 0.16 ^a^	-	3.97 ± 0.68 ^ab^

**Table 2 metabolites-09-00294-t002:** Growth responses and photosynthetic pigments in the two *Limonium* species after 30 days of treatment with the indicated NaCl concentrations. Mean ± SE values are shown (*n* = 5). Same letters within each column indicate homogeneous groups between treatments for each species according to the Tukey test (*p* < 0.05). *Leaf FW and root FW were expressed as percentages of the average weights in control plants, taken as 100%, with absolutes values of 6.08 g and 1.49 g (for *L. albuferae*) and 7.15 g and 1.22 g (for *L. dufourii*), respectively.

Parameters	NaCl (mM)	*L. albuferae*	*L. dufourii*
Leaf Number	0	13.4 ± 0.24 ^c^	24 ± 2.83 ^d^
	200	12.4 ± 0.81 ^c^	20.4 ± 1.14 ^c^
	400	12 ± 0.77 ^c^	18.6 ± 2.19 ^bc^
	600	9.8 ± 0.58 ^b^	16.2 ± 3.56 ^b^
	800	7.4 ± 0.24 ^a^	12.4 ± 2.19 ^a^
Leaf FW* (% of control)	0	100 ± 0.60 ^c^	100 ± 0.82 ^c^
	200	106.27 ± 7.91 ^c^	46.94 ± 3.28 ^b^
	400	83.26 ± 6.42 ^b^	37.72 ± 1.50 ^ab^
	600	49.93 ± 2.55 ^a^	31.65 ± 7.59 ^a^
	800	36.86 ± 1.76 ^a^	29.84 ± 3.40 ^a^
Roots FW* (% of control)	0	100 ± 0.18 ^a^	100 ± 0.38 ^a^
	200	204.11 ± 36.73 ^b^	240.50 ± 29.61 ^c^
	400	187.98 ± 18.56 ^b^	214.38 ± 17.13 ^bc^
	600	141.50 ± 34.02 ^ab^	159.88 ± 35.61 ^ab^
	800	108.72 ± 13.96 ^a^	142.12 ± 14.15 ^a^
WC % in Leaves	0	85.22 ± 0.54 ^e^	87 ± 0.30 ^c^
	200	82.46 ± 0.10 ^d^	84.77± 0.32 ^c^
	400	79.45 ± 0.54 ^c^	79.50 ± 0.92 ^b^
	600	75.82 ± 0.5 ^b^	75.54 ± 2.55 ^a^
	800	72.45 ± 0.55 ^a^	76.04 ± 0.65 ^ab^
WC% in Roots	0	77.73 ± 0.64 ^a^	83.75 ± 4.37 ^a^
	200	79.14 ± 0.70 ^a^	78.04 ± 9.73 ^a^
	400	78.42 ± 0.57 ^a^	79.60 ± 5.83 ^a^
	600	77.96 ± 0.63 ^a^	72.81 ± 5.67 ^a^
	800	76.80 ± 0.36 ^a^	75.47 ± 6.36 ^a^
Chl a (mg g ^−1^ DW)	0	4.99 ± 0.69 ^a^	3.61 ± 0.62 ^b^
	200	3.48 ± 0.13 ^a^	2.00 ± 0.30 ^a^
	400	2.35 ± 0.44 ^a^	1.96 ± 0.34 ^a^
	600	2.47 ± 0.32 ^a^	1.64 ± 0.26 ^a^
	800	3.22 ± 0.76 ^a^	1.64 ± 0.31 ^a^
Chl b (mg g ^−1^ DW)	0	3.56 ± 0.97 ^b^	2.06 ± 0.29 ^c^
	200	1.50 ± 0.34 ^a^	1.42 ± 0.25 ^b^
	400	0.81 ± 0.21 ^a^	0.95 ± 0.13 ^ab^
	600	0.76 ± 0.16 ^a^	0.66 ± 0.13 ^a^
	800	1.00 ± 0.26 ^a^	0.65 ± 0.06 ^a^
Caro (mg g ^−1^ DW)	0	1.76 ± 0.52 ^b^	1.04 ± 0.12 ^c^
	200	0.95 ± 0.30 ^a^	0.88 ± 0.07 ^bc^
	400	0.57 ± 0.09 ^a^	0.71 ± 0.06 ^ab^
	600	0.63 ± 0.15 ^a^	0.69 ± 0.08 ^ab^
	800	0.91 ± 0.23 ^a^	0.56 ± 0.04 ^a^

**Table 3 metabolites-09-00294-t003:** Ion contents in the two *Limonium* species after 30 days of treatments with the indicated NaCl concentrations. Mean ± SE values are shown (*n* = 5). Same letters within each column indicate homogeneous groups between treatments for each ion and species according to the Tukey test (*p* < 0.05).

Parameters	NaCl (mM)	Species	
		*L. albuferae*	*L. dufourii*
Na^+^ in roots (µmol g^−1^ DW)	0	150.26 ± 19.54 ^a^	325.60 ± 71.19 ^a^
	200	861.19 ± 71.59 ^b^	1104.48 ± 108.85 ^b^
	400	1081.68 ± 87.38 ^b^	1693.58 ± 119.04 ^c^
	600	1169.40 ± 97.91 ^b^	1744.62 ± 119.59 ^c^
	800	2186.31 ± 272.50 ^c^	2575.18 ± 63.85 ^d^
Na^+^ in leaves (µmol g^−1^ DW)	0	499.48 ± 31.68 ^a^	275.66 ± 68.10 ^a^
	200	2170.93 ± 121.15 ^b^	1990.42 ± 426.87 ^b^
	400	2644.25 ± 70.25 ^c^	2989.64 ± 160.04 ^c^
	600	2845.45 ± 308.40 ^c^	3042.36 ± 199.10 ^c^
	800	3033.51 ± 115.87 ^c^	4271.14 ± 135.61 ^d^
Cl^−^ in roots (µmol g^−1^ DW)	0	110.208 ± 19.83 ^a^	123.51 ± 28.55 ^a^
	200	766.74 ± 75.35 ^b^	646.08 ± 85.31 ^b^
	400	1025.84 ± 83.21 ^c^	892.34 ± 55.12 ^c^
	600	1383.95 ± 72.81 ^d^	1120.20 ± 83.55 ^d^
	800	1575.37 ± 55.61 ^d^	1473.89 ± 111.90 ^e^
Cl^−^ in leaves (µmol g^−1^ DW)	0	437.95 ± 32.76 ^a^	671.27 ± 31.86 ^a^
	200	1120.56 ± 67.86 ^b^	1530.25 ± 74.81 ^b^
	400	1451.80 ± 35.61 ^c^	1723.05 ± 32.51 ^b^
	600	1508.57 ± 137.70 ^c^	1746.18 ± 110.38 ^b^
	800	1689.12 ± 65.28 ^c^	2472.19 ± 75.76 ^c^
K^+^ in roots (µmol g^−1^ DW)	0	180.38 ± 31.32 ^a^	149.00 ± 18.31 ^b^
	200	222.38 ± 26.56 ^a^	93.85 ± 24.03 ^ab^
	400	172.02 ± 26.04 ^a^	64.62 ± 12.72 ^a^
	600	177.11 ± 11.67 ^a^	78.94 ± 17.08 ^a^
	800	170.94 ± 16.49 ^a^	146.49 ± 9.15 ^b^
K^+^ in leaves (µmol g^−1^ DW)	0	607.93 ± 31.41 ^b^	774.63 ± 78.57 ^b^
	200	410.06 ± 25.32 ^a^	490.56 ± 30.28 ^a^
	400	458.93 ± 19.28 ^a^	494.16 ± 38.64 ^a^
	600	436.83 ± 35.94 ^a^	533.84 ± 81.54 ^a^
	800	414.70 ± 6.06 ^a^	444.43 ± 32.07 ^a^
Ca^2+^ in roots (µmol g^−1^ DW)	0	30.16 ± 2.14 ^a^	43.00 ± 9.44 ^a^
	200	106.50 ± 9.37 ^b^	84.50 ± 8.96 ^b^
	400	121.91 ± 13.30 ^bc^	130.32 ± 5.31 ^c^
	600	145.20 ± 14.16 ^c^	129.29 ± 3.73 ^c^
	800	191.83 ± 5.00 ^d^	176.31 ± 4.11 ^d^
Ca^2+^ in leaves (µmol g^−1^ DW)	0	249.57 ± 13.90 ^a^	210.18 ± 14.96 ^a^
	200	305.30 ± 15.32 ^ab^	322.30 ± 24.60 ^b^
	400	360.52 ± 18.77 ^b^	394.36 ± 23.20 ^b^
	600	302.40 ± 30.14 ^ab^	341.84 ± 10.12 ^b^
	800	353.95 ± 19.94 ^b^	349.03 ± 38.56 ^b^

**Table 4 metabolites-09-00294-t004:** Factorial ANOVA (F values) considering the effect of Species (A), Treatment (B), Organ (C), and their interactions (A × B; A × C; B × C; A × B × C) on growth parameters (leaf number, FW, WC%), photosynthetic pigments (Chl a, Chl b, Caro) and ions (Na^+^, K^+^, Cl^−^, Ca^2+^) in *L. albuferae* and *L. dufourii*.

Parameter	A (Species)	B (Treatment)	C (Organ)	A × B Interaction	A × C Interaction	B × C Interaction	A × B × C Interaction
Leaf (number)	0.000 ***	0.498	-	0.004 **	-	-	-
FW	0.000 ***	0.007 **	0.000 ***	0.002 **	0.000 ***	0.081 ^ns^	0.032 *
WC%	0.001 **	0.631 ^ns^	0.407 ^ns^	0.471 ^ns^	0.382 ^ns^	0.586 ^ns^	0.466 ^ns^
Chl a	0.062 ^ns^	0.030 *	-	0.533 ^ns^	-	-	-
Chl b	0.000 ***	0.180 ^ns^	-	0.350 ^ns^	-	-	-
Caro	0.002 **	0.098 ^ns^	-	0.006 **	-	-	-
Na^+^	0.000 ***	0.000 ***	0.000 ***	0.010 **	0.000 ***	0.481 ^ns^	0.039 *
K^+^	0.000 ***	0.899 ^ns^	0.242 ^ns^	0.241 ^ns^	0.000 ***	0.000 ***	0.465 ^ns^
Cl^−^	0.000 ***	0.000 ***	0.000 ***	0.039 *	0.018 *	0.000 ***	0.026 *
Ca^2+^	0.000 ***	0.862 ^ns^	0.000 ***	0.088 ^ns^	0.010 *	0.326 ^ns^	0.259 ^ns^

*, **, *** significant at *p* = 0.05, 0.01 and 0.001 respectively; ns: not significant.

**Table 5 metabolites-09-00294-t005:** Factorial ANOVA (F values) considering the effect of Species (S), Treatment (T), and their interactions (S × T) on metabolic profiles of leaf carbohydrates, phosphoric and organic acids and amino acids in *L. albuferae* and *L. dufourii.*

Group of Compounds	Parameter	S	T	S × T
Carbohydrates	Erythritol	0.0099 **	0.0000 ***	0.0419 **
	Fructose	0.0000 ***	0.0408 **	0.0660 *
	Glucose	0.6060 ^ns^	0.0082 **	0.2290 ^ns^
	Glycerol	0.1200 ^ns^	0.4000 ^ns^	0.2700 ^ns^
	Myoinositol	0.5321 ^ns^	0.0109 *	0.8501 ^ns^
	Raffinose	0.2707 ^ns^	0.0000 ***	0.0097 **
	Rhamnose	0.0191 *	0.0000 ***	0.6961 ^ns^
	Sucrose	0.0240 *	0.0000 ***	0.0650 ^ns^
Inorganic Acids	Phosphoric acid	0.3124 ^ns^	0.4720 ^ns^	0.2133 ^ns^
Organic Acids	Citric acid	0.0044 **	0.0907 ^ns^	0.0303 *
	Glyceric acid	0.3516 ^ns^	0.0000 ***	0.0210 *
	Maleic acid	0.2020 ^ns^	0.0840 ^ns^	0.3300 ^ns^
	Malic acid	0.0737 ^ns^	0.0533 ^ns^	0.0201 *
	Succinic acid	0.6010 ^ns^	0.5586 ^ns^	0.0064 **
	Threonic acid	0.0018 **	0.0000 ***	0.0754 ^ns^
Amino Acids	Alanine	0.6222 ^ns^	0.0034 **	0.2083 ^ns^
	Asparagine	0.9672 ^ns^	0.0322 *	0.0053 **
	Aspartic acid	0.1198 ^ns^	0.0064 **	0.0039 **
	GABA	0.2916 ^ns^	0.4754 ^ns^	0.1937 ^ns^
	Glutamic acid	0.0000 ***	0.5378 ^ns^	0.5780 ^ns^
	Glutamine	0.0000 ***	0.0000 ***	0.0000 ***
	Glycine	0.7987 ^ns^	0.0669 ^ns^	0.0084 **
	Isoleucine	0.0115 *	0.0009 ***	0.0343 *
	Leucine	0.4357 ^ns^	0.0024 **	0.0102 *
	Lysine	0.0335 *	0.0001 ***	0.0975 ^ns^
	Phenylalanine	0.9134 ^ns^	0.0000 ***	0.0000 ***
	Proline	0.2250 ^ns^	0.0000 ***	0.0353 *
	Pyroglutamic acid	0.0000 ***	0.0003 **	0.0003 **
	Serine	0.6344 ^ns^	0.0006 ***	0.1424 ^ns^
	Threonine	0.0002 **	0.0000 ***	0.0564 ^ns^
	Tryptophan	0.1810 ^ns^	0.0170 *	0.0353 *
	Valine	0.0422 *	0.0000 ***	0.0106 *

*, **, *** significant at *p* = 0.05, 0.01 and 0.001 respectively; ns: not significant.

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
