# Peer review of "Insights on Salt Tolerance of Two Endemic *Limonium* Species from Spain"

_metabolites, 2019, doi:10.3390/metabo9120294_

Round 1
Reviewer 1 Report
This paper reports the physiological and metabolomic responses of two Limonium species, L. albuferae and L. doufourii, exposed to salt stresses. This paper includes some novel and interesting findings in salt stress responses of the plants. The following issues, however, should be appropriately addressed before publication.
First, I understand that the metabolite contents are relative values. However, it is necessary to display not only relative values within the same plant species but also relative values between the two types of plants. Otherwise, the comparison between the salt tolerance mechanisms between the two species described in Discussion is not possible.
L42, This is not true. There are lots of families containing halophyte plants.
Introduction section is redundant and needs reconsideration. For example, It has been mentioned many times that the Limonium species are in danger of extinction.
L67 and 81, triploid species (2n = 27) and triploid (2n = 26), Are these notations correct?
L101, Because the germination ability was not lost, so ‘recover’ is not appropriate.
Figure legends are not fully explained. Especially, Fig.2 needs plant age and number of days after treatment. Fig.4 needs the information that it is a metabolite of the leaf and the graph shows the relative contents.
L253-254, This is not true.
L289 and so on. The data do not represent ‘contents’ but ‘relative contents’.
Discussion section, especially L354-L391, is redundant and should be brief.
L418-421, This is not true. Most halophytes including monocot plats accumulate sodium and chloride ions in shoots.
L443, Two species used in this study are salt-secretion types? Otherwise, this description has no meaning.
Some notations in the text should be unified. For example, both proline and Pro are used in the same paragraph. Taxa and species.
English should be corrected throughout manuscript. For example, in2 line 81, ‘apomictic recently described species’, in line 167, ‘root FW not only did not decrease’, in line 192, ‘higher levels that Cl-’, etc.
In Table2,
I'm wondering if the statistical process is correct. Especially Chl a in L. albuferae.
WC in shoot is missing.
Author Response
Rev. 1
This paper reports the physiological and metabolomic responses of two Limonium species, L. albuferae and L. dufourii, exposed to salt stresses. This paper includes some novel and interesting findings in salt stress responses of the plants.
- Thank you for your positive comment on our manuscript
The following issues, however, should be appropriately addressed before publication.
First, I understand that the metabolite contents are relative values. However, it is necessary to display not only relative values within the same plant species but also relative values between the two types of plants. Otherwise, the comparison between the salt tolerance mechanisms between the two species described in Discussion is not possible.
- The reviewer is right. In the revised version, an additional column has been included in Table S1, with the ratios between mean values registered in the two species for each detected metabolite, supporting the comments in Discussion on the comparison of salt tolerance mechanisms in the two species.
L42, This is not true. There are lots of families containing halophyte plants.
- We agree with the reviewer’s comment, and have modified this sentence accordingly
Introduction section is redundant and needs reconsideration. For example, It has been mentioned many times that the Limonium species are in danger of extinction.
- The’ Introduction’ section has been extensively modified in the revised version, following the reviewer’s suggestion. Redundant and non-essential information has been deleted, mostly referring to some morphological and ecological characteristics of the investigated species and their classification as endangered species. The overall length of this section may not have been reduced since we have added some information on general mechanisms of salt tolerance in plants, following the recommendations of reviewer 2.
L67 and 81, triploid species (2n = 27) and triploid (2n = 26), Are these notations correct?
- They are indeed correct, according to the references (cited in the text) reporting the chromosome numbers of the two species.
L101, Because the germination ability was not lost, so ‘recover’ is not appropriate.
- This is true, and we have replaced ‘recover’ by ‘maintain’ in this sentence in the text. However, assays in which seeds that have not germinated in the presence of salt are washed and then germinated in water, are always defined as ‘germination recovery’ or ‘recovery of germination’ assays, and we have maintained this terminology in Figure 1, Table 1 and in the description of these result in section 2.1. Seed germination.
Figure legends are not fully explained. Especially, Fig.2 needs plant age and number of days after treatment. Fig.4 needs the information that it is a metabolite of the leaf and the graph shows the relative contents.
- We thank the reviewer for pointing out that some figures were not well explained. We have carefully checked the legends of figures and tables to include, where necessary, additional information so that they can be understood without reference to the text.
L253-254, This is not true.
- Thank you for detecting the error. In control, non-stressed plants, lysine was indeed detected only in Limonium dufourii, not in L. albuferae. This has been corrected in the revised version.
L289 and so on. The data do not represent ‘contents’ but ‘relative contents’.
- Yes, indeed. In the revised version we refer to ‘relative contents’ in the appropriate sections, throughout the text and also in figures ant tables.
Discussion section, especially L354-L391, is redundant and should be brief.
- We have deleted several paragraphs of the Discussion, containing non-essential information, following the reviewer’s comment, substantially reducing the original text.
L418-421, This is not true. Most halophytes including monocot plats accumulate sodium and chloride ions in shoots.
- This is true, although the highest ion concentrations in the plants aerial parts are generally reached in dicotyledonous halophytes, especially in succulent plants. Nevertheless, we agree that the original text could be misleading, and it has been modified in the revised version, to state clearly that sodium and chloride accumulation in shoots is a specific trait of halophytes, although generally more efficient in dicotyledonous species.
L443, Two species used in this study are salt-secretion types? Otherwise, this description has no meaning.
- They are, indeed, but the reviewer’s comment made us realise that this was not explicitly mentioned in the manuscript with reference to the two investigated species. We have highlighted this in text of the revised version.
Some notations in the text should be unified. For example, both proline and Pro are used in the same paragraph. Taxa and species.
- We use the standard abbreviation of the amino acid, Pro, throughout the text except at first mention in the Abstract. We have also unified the reference to ‘species’, not ‘taxa’.
English should be corrected throughout manuscript. For example, in2 line 81, ‘apomictic recently described species’, in line 167, ‘root FW not only did not decrease’, in line 192, ‘higher levels that Cl-’, etc.
- The English grammar and style has been carefully checked throughout the manuscript, correcting all detected mistakes, including those pointed out by the reviewer
In Table2,I'm wondering if the statistical process is correct. Especially Chl a in L. albuferae.
- The statistic analysis is correct, but the legend was wrong as we calculated mean values ± Standard Error (SE) but ‘Standard Deviation (SD)’ was mistakenly written in the legend; this has been corrected in the revised version. In addition, thanks to the reviewer’s observation, we were aware that we had used SD for the germination time data (Table 1), whereas for all other experiments mean values are followed by SE. To present the data uniformly, we have recalculated the statistical error of the germination time values and modified Table 1, showing now ‘Means ± SE’ instead of ‘Means ± SD’.
WC in shoot is missing.
- We apologise for the mistake. The missing data have been included in the revised version of Table 2.
Reviewer 2 Report
The manuscript is based on the presentation of the result of extensive analysis of various morphological, physiological and biochemical parameters of two endemic Limonium Species under salinity. The authors set a goal - to detect differences in salinity resistance of two plant species and try to explain differences by metabolic profiling data. The manuscript is within Aims and Scope of Journal. The presented article is interest, has a new data and it may be published after significant revision.
Questions to the authors
When studying the profile of metabolites, the authors studied only carbohydrates, polyols, organic acids and amino acids. Why was the choice made in favor of these compounds? It is known that in response to salt stress, plants synthesize many protective substances, for example antioxidants (phenols, ascorbic acid and glutathione). The study of a wider range of substances would enhance the scientific significance of the study. An explanation should be presented in Introduction section. The analysis of physiological parameters (FW, DW, WC) is made for both roots and leaves. Why there was no study of the metabolic profile for the roots, stress-dependent processes in which are no less interesting than in the leaves. The data in table S1 are given in relative units. In order to assess the share of a particular osmolyte in the adaptation of a plant to salinity, it is necessary to have quantitative data (umol/g, %, ppm). Otherwise, the conclusions about major osmolyte cannot be verified.
Remarks
The introduction contains an unnecessarily detailed description of the ecological and genetic characteristics of the studied species. I believe that this part should be significantly reduced. Instead, you can add information to the mechanisms of salt tolerance, some of which are studied in this article. Fig. 1. Error designations should be made no higher than 100%. Table 2. Missing signature WC% in leaves Table 3. Decipher what the signs ‘and * mean? Fig. 3. The description of the picture is not clear. On which part should I look for the leaves and roots of L. Albuferae or L. Dufourii? Parts (a) and (c) are representing the total weight of the plants? Myo-Inositol and erythritol and glycerin are Polyol, not carbohydrates. Is phosphoric acid organic? (line 268) In the signatures of figures and tables (including S1) relating to metabolites should indicate that all data for plant leaves. Figure 6. Designations in the caption of the figure are not in the figure itself (LA), (LD), (C). In the Methods section, it is not indicated how the extraction of calcium was carried out. A significant portion of the calcium reserves may be in the form of insoluble compounds with cell wall and salts. If it was determined in samples for Na and K, then it should be indicated as bioavailable calcium.Author Response
The manuscript is based on the presentation of the result of extensive analysis of various morphological, physiological and biochemical parameters of two endemic Limonium Species under salinity. The authors set a goal - to detect differences in salinity resistance of two plant species and try to explain differences by metabolic profiling data. The manuscript is within Aims and Scope of Journal. The presented article is interest, has a new data and it may be published after significant revision.
- We thank the reviewer for considering our work as an interesting contribution
Questions to the authors
When studying the profile of metabolites, the authors studied only carbohydrates, polyols, organic acids and amino acids. Why was the choice made in favor of these compounds? It is known that in response to salt stress, plants synthesize many protective substances, for example antioxidants (phenols, ascorbic acid and glutathione). The study of a wider range of substances would enhance the scientific significance of the study. An explanation should be presented in Introduction section. The analysis of physiological parameters (FW, DW, WC) is made for both roots and leaves. Why there was no study of the metabolic profile for the roots, stress-dependent processes in which are no less interesting than in the leaves.
- We fully agree with the reviewer in that a wider metabolomics analysis, including secondary metabolites, such as volatiles and antioxidant phenolics, performed in both, roots and leaves, could provide interesting additional information regarding the mechanisms of salt tolerance in the investigated Limonium species, but this is outside the scope of the present manuscript. This is an ongoing project, and work is in progress to assess, for example, the relative degree of oxidative stress affecting salt-treated plants of both species (by determining MDA and H2O2 levels) and the possible activation of enzymatic and non-enzymatic antioxidant systems; in this context, and depending on the initial results, we will consider to perform targeted metabolomics analyses of, for example, phenolic compounds. For the time being, we have only carried out the untargeted analysis of primary metabolites described here. Some comments have been included in the Introduction and Materials and Methods to better explain this approach.
The data in table S1 are given in relative units. In order to assess the share of a particular osmolyte in the adaptation of a plant to salinity, it is necessary to have quantitative data (umol/g, %, ppm). Otherwise, the conclusions about major osmolyte cannot be verified.
- It is true; all data are expressed as relative contents of the detected metabolites, which is the way the Metabolomics Service of the Institute for Molecular and Cell Biology, Polytechnic University of Valencia, routinely provides the results of untargeted metabolomics analyses. They consider quantification, using specific standards, only for targeted analyses. In any case, also answering to a first reviewer’s comment, in the revised version of Table S1 an additional column has been included, with the ratios between mean relative values registered in the two species for each detected metabolite. We think that presenting these data relating the two species is sufficient to cover the aim of our comparative study.
The introduction contains an unnecessarily detailed description of the ecological and genetic characteristics of the studied species. I believe that this part should be significantly reduced. Instead, you can add information to the mechanisms of salt tolerance, some of which are studied in this article.
- We fully agree with the reviewer. Detailed descriptions on the topics mentioned in the above comment have been reduced or completely deleted from the Introduction. We have also added a new paragraph with general information on the main mechanisms of salt tolerance in plants.
Fig. 1. Error designations should be made no higher than 100%.
- It appeared like that because in the original figure we used standard deviations ‘SD’ of the means for the statistical analysis of the germination data. By using instead standard errors ‘SE’ in the present version, error designations are under 100%.
Table 2. Missing signature WC% in leaves
- We apologise for this mistake, which has been corrected in the revised version of Table 2
Table 3. Decipher what the signs ‘and * mean?
- Sorry for the mistake. “’”¨is just a typing error, and “*” is a leftover from a previous version, before submission of the manuscript, representing the outcome of a Student test comparing the ion contents in the two species for each treatment. Finally, we opted for a one-way ANOVA, which is much more explanatory. These symbols have been deleted in the new version of Table 3.
Fig. 3. The description of the picture is not clear. On which part should I look for the leaves and roots of L. Albuferae or L. Dufourii? Parts (a) and (c) are representing the total weight of the plants?
- Figure 3 is the graphic representation of the outcome of the factorial ANOVA, first relating the species with the treatment (without differentiating organs) (panels a, c), and second relating the treatment with the organ (mixing the values of the two species) (panels b, d). We admit that the description of this figure was not clear enough in the original version, and have modified the corresponding text in ‘Results’ and the legend, to better explain what is shown in Fig. 3.
Myo-Inositol and erythritol and glycerin are Polyol, not carbohydrates.
- Myo-inositol, erythritol and glycerol are not sugars, but polyols (or sugar-alcohols). They are generally considered as belonging to the group of ‘carbohydrates’, even though for some of them the hydrogen–oxygen atom ratio may not be 2:1 (as in water); for example, the chemical formula of erythritol is C4H10O4 but that of inositol is C6H12O6. In the text and in figures and tables, to simplify, we have generally maintained the term ‘carbohydrates’ to refer to this group of metabolites, although we also use ‘sugar and polyols’ as an equivalent.
Is phosphoric acid organic? (line 268)
- Of course not. Thank you for pointing out this basic mistake, which we have corrected in the text, figures and tables in the revised version, referring separately to ‘phosphoric acid’ and ‘organic acids’.
In the signatures of figures and tables (including S1) relating to metabolites should indicate that all data for plant leaves.
- Included in the revised version
Figure 6. Designations in the caption of the figure are not in the figure itself (LA), (LD), (C).
- Figure 6 is the Loading scatter plot of the PLS analysis, showing the distribution of the analysed parameters (metabolites), not that of the samples that appear in Figure 5. However, in the original legend, we mistakenly included references to the two species and the different treatments, which did not belong there, making the legend extremely confusing; we apologise for that. We have modified the legend accordingly in the revised version.
In the Methods section, it is not indicated how the extraction of calcium was carried out. A significant portion of the calcium reserves may be in the form of insoluble compounds with cell wall and salts. If it was determined in samples for Na and K, then it should be indicated as If it was determined in samples for Na and K, then it should be indicated as bioavailable calcium.
- All ions were determined in the same extracts, prepared by boiling the plant material in water, as it is described in Methods. We prefer aqueous extraction since we consider that determination of soluble ions is physiologically more relevant than measuring ‘total’ ion levels after extraction under strong acidic conditions – an alternative method often used for ions extraction from plants. Therefore, the reviewer is completely right in that insoluble calcium present, for example, in cell walls, will not be extracted under these conditions. We have followed his/her suggestion, using the term ‘bioavailable calcium’ in the description of the ion extraction and determination procedures.
Reviewer 3 Report
This manuscript could be an interesting contribution to the knowledge of the studied species in a climate change scenario, however:
1-the authors present an extensive amount of data that is not properly discussed;
2-a general aim is not referred, only specific objectives;
3-the discussion chapter becomes quite long not with the data obtained but with the state of the art about the halophyte physiology in general and very little on the species under study.
4-very little is proposed about the management programs of these two endangered species
Specific remarks:
Figure and table captions should be more succinct and accurate
Line 63: “et al.” must be replaced by the authors name
Lines 189, 190, 191 and 197: species and genus name in italics
Author Response
This manuscript could be an interesting contribution to the knowledge of the studied species in a climate change scenario, however
- Thank you for considering our work of interest
1-the authors present an extensive amount of data that is not properly discussed;
- The Discussion has been extensively modified, also following the other reviewers’ suggestions. We believe that the discussion of the data has been substantially improved in the revised version.
2-a general aim is not referred, only specific objectives;
- We completely agree with this comment, and have modified the last paragraph of the Introduction to include the description of the general aims of the work.
3-the discussion chapter becomes quite long not with the data obtained but with the state of the art about the halophyte physiology in general and very little on the species under study.
- Many general comments on salt tolerance mechanisms and the physiology of halophytes have been reduced or completely deleted from the Discussion and some are now included in the Introduction, which was also suggested by the third reviewer. What is not possible is to include more information on the selected Limonium species since this is the first study on the physiological and biochemical responses to salt stress of the two endemic species, and there are no data reported in the literature.
4-very little is proposed about the management programs of these two endangered species
- A new subsection has been included at the end of the Discussion, addressing the point raised by the reviewer, and proposing specific strategies for the implementation of conservation and reintroduction programmes of these threatened endemic species, based on the results presented in the paper..
Specific remarks:
Figure and table captions should be more succinct and accurate
- We could not reduce the legends, as they should contain enough information to make figures and tables understandable without reference to the text. However, we must admit that some captions were unclear or inaccurate, and we have modified them in the revised version.
Line 63: “et al.” must be replaced by the authors name
- Done
Lines 189, 190, 191 and 197: species and genus name in italics
- Corrected
Round 2
Reviewer 3 Report
Overall the authors updated satisfactorily the proposed revisions. However, some corrections must be done:
Line 47-“ thesespecies” must be replaced by -“ these species”
Line 70-“ apomictic” must be replaced by -“ as apomictic”
Line 169-“ young of plants” must be replaced by -“ young plants
Line 243-“ Root t” must be replaced by -“ Root”
Line 386-“ speciesof” must be replaced by -“ species of”
Author Response
Thank you very much for your detailed revision of the manuscript. All errors were corrected.